# Germinal center trajectories and transcriptional signatures define CLL subtypes and their pathway regulators

Ahmed Mohamed[1]*, Luca Giudice[1], José Basílio[2], Sabina Barresi[3,4], Pradeep Kumar Kopparapu[5], Vanda Friman[6], Marco Tartaglia[3], Tarja Malm[1], Andreas Tilevik[7], Ola Grimsholm[2,5]*

1 Faculty of Health Sciences, A.I. Virtanen Institute for Molecular Sciences, University of Eastern Finland, Kuopio, Finland, 2 Department of Pathophysiology and Allergy Research, Centre for Pathophysiology, Infectiology and Immunology, Medical University Vienna, Vienna, Austria, 3 Molecular Genetics and Functional Genomics, Bambino Gesù Children's Hospital, IRCCS, Rome, Italy, 4 Current Address: Pathology Unit, Bambino Gesù Children's Hospital, IRCCS, Rome, Italy, 5 Department of Rheumatology and Inflammation Research, The Sahlgrenska Academy, University of Gothenburg, Gothenburg, Sweden, 6 Department of Infectious Diseases, Institute of Biomedicine, The Sahlgrenska Academy, University of Gothenburg, Gothenburg, Sweden, 7 School of Bioscience, University of Skövde, Skövde, Sweden

* ahmed.mohamed@uef.fi (AM); ola.grimsholm@meduniwien.ac.at (OG)

## Abstract

Chronic lymphocytic leukemia (CLL) is divided into unmutated (UM-CLL) and mutated (M-CLL) subtypes depending on somatic hypermutation (SHM) frequency in their immunoglobulin heavy chain V (IGHV) region. We previously demonstrated that CD27bright memory B cells (MBCs) are germinal center (GC)-dependent with higher mutation rate, whereas CD27dull MBCs accumulate fewer mutations and originate independently from the GC. We conducted a meta-transcriptomic analysis on bulk RNA data from 116 individuals combining four CLL cohorts and healthy B cell subsets (naïve, CD27dull and CD27bright MBCs) to decipher the transcriptional and mechanistic functions of CLL subtypes. CD27bright MBCs showed more transcriptional similarity to M-CLL rather than UM-CLL. Functional enrichment analysis revealed that *LPL*, *ZNF667* and *ZNF667-AS1* are potential informative biomarkers for stratification of CLL subtypes. They are part of the mechanistic regulatory pathways of CLL pathology through cholesterol and Epithelial Mesenchymal Transition (EMT) regulation. We applied markers for the GC B-cell substages to map *in silico* the CLL cohorts to their potential GC B cell counterpart. UM-CLL represented transcriptional mimicry to an early intermediary GC substage whereas M-CLL mimicked later substages in the GC. This could potentially explain the IGHV mutational status of M-CLL as well as hypothesize that CLL subtypes could derive from a GC-dependent pathway.

**Data availability statement:** Public data was used in this study and details are in S1 File. All data are contained within the paper and the analysis pipeline is available on Zenodo (https://doi.org/10.5281/zenodo.15000801). The raw data of the RNA sequencing files from anonymous healthy donors' buffy coats used in this study have been uploaded to NCBI (https://www.ncbi.nlm.nih.gov/) with series number GSE307095.

**Funding:** OG: Assar Gabrielsson foundation, Anna-Lisa and Bror Björnsson foundation, Adlerbert Research Foundation, the Royal Sciences and Arts in Gothenburg and the Austrian Science Fund (FWF), project PAT3959823. MT: Associazione Italiana per la Ricerca sul Cancro (IG28768). AM: European Union's Horizon 2020 research and innovation programme under the Marie Skłodowska-Curie agreement No101034307. The funders had no role in study design, data collection and analysis, decision to publish, or preparation of the manuscript.

**Competing interests:** The authors have declared that no competing interests exist.

**Abbreviations:** BCR, B Cell Receptor; CLL, Chronic Lymphocytic Leukemia; *CLLU1*, Chronic Lymphocytic Leukemia Up-Regulated 1; *COBLL1*, Cordon-Bleu WH2 Repeat Protein Like 1; *CRY1*, Cryptochrome Circadian Regulator 1; DEGs, Differentially Expressed Genes; DZ, Dark Zone; EMT, Epithelial Mesenchymal Transition; GC, Germinal Center; GSVA, Gene Set Variation Analysis; IGHV, Immunoglobulin Heavy chain Variable region; INT, Intermediary zone; *KLHL6*, Kelch Like Family Member 6; logFC, log Fold-Change; *LPL*, Lipoprotein Lipase; LZ, Light Zone; MA, Mean Difference; MBC, Memory B Cell; M-CLL, Mutated Chronic Lymphocytic Leukemia; MDS, Multi-dimensional Scaling; MET, Mesenchymal to Epithelial Transition; PBL, Plasmablast; PBMCs, Peripheral Blood Mononuclear Cells; *PHEX*, Phosphate Regulating Endopeptidase X-Linked; PreM, Pre-Memory B Cell; *S100A4*, S100 Calcium Binding Protein A4; *SEPT10*, Septin 10; SHM, Somatic Hypermutation; UM-CLL, Unmutated Chronic Lymphocytic Leukemia; *ZAP70*, Zeta-Associated Protein-70; *ZNF667*, Zinc Finger protein 667; *ZNF667-AS1*, Zinc Finger protein 667 Anti-Sense-1

## Introduction

Chronic lymphocytic leukemia (CLL) is the most common leukemia in the elderly population and remains in many cases an incurable disease [1]. CLL can be divided into two main subtypes defined by the mutation status of the immunoglobulin heavy chain V segment (IGHV), where the categorical cutoff of somatic hypermutation (SHM) in the IGHV locus is ≤ 2.0% for the unmutated (UM-CLL) subtype and >2.0% for the mutated (M-CLL) subtype [2,3]. The cellular origin of the two CLL subtypes is still debated, and the identification of their healthy counterparts is a current challenge [4]. Different hypotheses on the normal counterpart have been suggested, such as naïve, memory or marginal zone-like B cells [4–7]. Furthermore, it has been proposed that UM-CLL would arise through a T-cell-independent pathway and that M-CLL would mainly stem from the germinal center (GC) reaction [8,9]. Another epigenetic study conducted by Oakes and colleagues proposed that CLL arises from a continuum of maturation stages of normal B cells [10].

Early studies on microarrays and RNA sequencing data identified Zeta-Associated Protein-70 (*ZAP70*) as a biomarker for CLL subtypes as well as key regulatory pathways such as B cell receptor (BCR) and Tumor Necrosis Factor signaling pathways [11–16]. This has affected the treatment options where BTK inhibitors are currently used to inhibit BCR signaling in CLL [17]. Other pathways are still not extensively studied in CLL. Among these, the neuroactive ligand receptor was originally described as an important group of neuroreceptors, but also oncogenes that contribute to various cancers such as lung cancer, colon cancer, prostate cancer, pancreatic cancer, glioblastoma and osteosarcoma [18–23].

Previous efforts to investigate the transcriptomic landscape of the two CLL subtypes compared to multiple control B-cell subtypes, mainly naïve and memory B cells (MBCs), have shown that UM-CLL is more similar to the mature naïve B cell, while M-CLL shows more resemblance to a memory rather than any other B cell subtype [7]. In a seminal study by Seifert and colleagues, they identified CD5+CD27- B cells as the lineage giving rise to the UM-CLL subtype, while CD5+CD27+ MBCs as the healthy counterpart of the M-CLL subtype [24]. However, it is not clear whether CD5 becomes expressed upon malignant transformation in CLL or if the precursor cell also expresses it [7].

We have previously shown that the pool of peripheral blood MBCs can be divided into two major populations defined by pan-memory B cell marker CD27 into CD27dull and CD27bright MBCs [25]. The CD27dull MBCs can be formed without T cells or GC, contain few somatic mutations and are characterized by a high proliferative potential, whereas CD27bright MBCs are highly somatically mutated, GC-dependent and quickly give rise to antibody-producing cells upon *in vitro* stimulation. Here, we set out to investigate the potential cellular origin of M- and UM-CLL subtypes using multiple CLL transcriptomic cohorts downloaded from public databases and compared with previously and newly (herein) generated RNA sequencing data from mature-naïve (CD24+CD38int) B cells, CD27dull MBCs (CD24hiCD27dull) and CD27bright MBCs (CD24hiCD27bright). We performed a transcriptomic meta-analysis to decipher

commonalities and differences between M- and UM-CLL subtypes against healthy B cells. Additionally, we used single-cell data from Holmes and colleagues describing multiple GC B cell stages to investigate the potential GC origin of the two CLL subtypes [26]. Our results pinpoint that M-CLL is more similar to CD27bright MBCs than the other B cell subtypes. UM-CLL diverges more than M-CLL from the healthy B cell subsets from the peripheral blood analyzed. Pathway analysis demonstrates that M- and UM-CLL shared many of the altered pathways when compared to CD27bright MBCs, including both annotated (cell-cell adhesion and regulation of cholesterol transport) and novel (neuroactive ligand-receptor inter-action) functional molecular sets. In addition, our analysis proposes Lipoprotein Lipase (*LPL*), Zinc Finger protein 667 (*ZNF667*) and Zinc Finger protein 667 Anti-Sense-1 (*ZNF667-AS1*) as informative biomarkers for CLL patient stratification. It also suggests that UM-CLL may have originated from an early intermediary GC B cell stage, which could explain its low IGHV mutation content. Conversely, M-CLL is likely to arise from a later stage of the GC.

## Results

### The M-CLL subtype is more closely related to healthy donor peripheral blood MBC subsets compared to the UM-CLL subtype

Two datasets from healthy donors were used, comprising nine individuals with three subsets of B cells (nine naïve B cells, nine CD27bright MBCs and four CD27dull MBCs) (S1 File and S1 Fig). Four CLL datasets were used, and 16 outliers were detected (10 M-CLL and 6 UM-CLL) and removed (S2 File). A total of 107 CLL patients with two subtypes (57 M-CLL and 50 UM-CLL) were finally used for the analysis (S1 File). Batch correction for both the run type and dataset origin after excluding outliers shows homogenous data (Figs 1A and 1B). MDS plot for the integrated datasets shows good homo-geneity in each of the five groups (Fig 1C). Differential expression analysis for each pairwise comparison and the lists of highly differentially expressed genes (DEGs) with both a significant adjusted p-value < 0.05 and an absolute log Fold-Change (logFC) > 1 are shown in S3 File.

Mean difference (MA) plot to visualize the significant highly expressed DEGs from the pairwise comparison mutated *vs* unmutated, shows a total of 94 DEGs; 17 upregulated and 77 downregulated DEGs, respectively (Fig 2A). MDS plot was used to visualize and investigate the importance of these 94 DEGs through dimensionality reduction (Fig 2B). MDS plot using these 94 significant DEGs for all the subjects, shows coherent clustering in each CLL subgroup and good separation between the CLL subgroups. The healthy subsets of naïve B cells and MBCs were adequately separated. It is noticeable that healthy subgroups cluster closer to the M-CLL subtype rather than the UM-CLL subtype, specifically the CD27bright subgroup. A comprehensive heatmap representing all 94 DEGs, shows five separate clusters where the

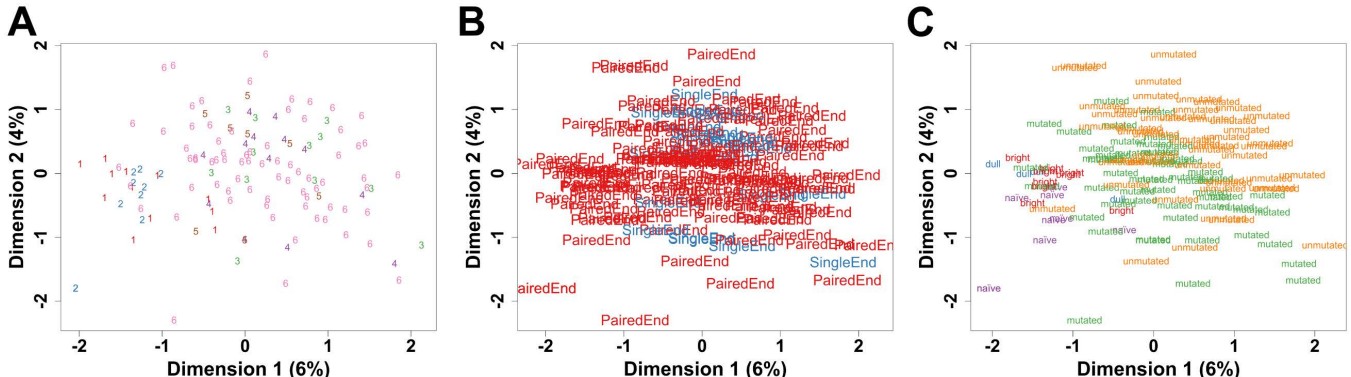

**Fig 1. Quality control for all six datasets (n = 116).** MDS plots after outlier removal and batch correction for both run type (paired or single end) and dataset origin, show homogenous integration of the datasets according to the origin of dataset **(A)**, run type **(B)** and the five groups in comparison **(C)**.

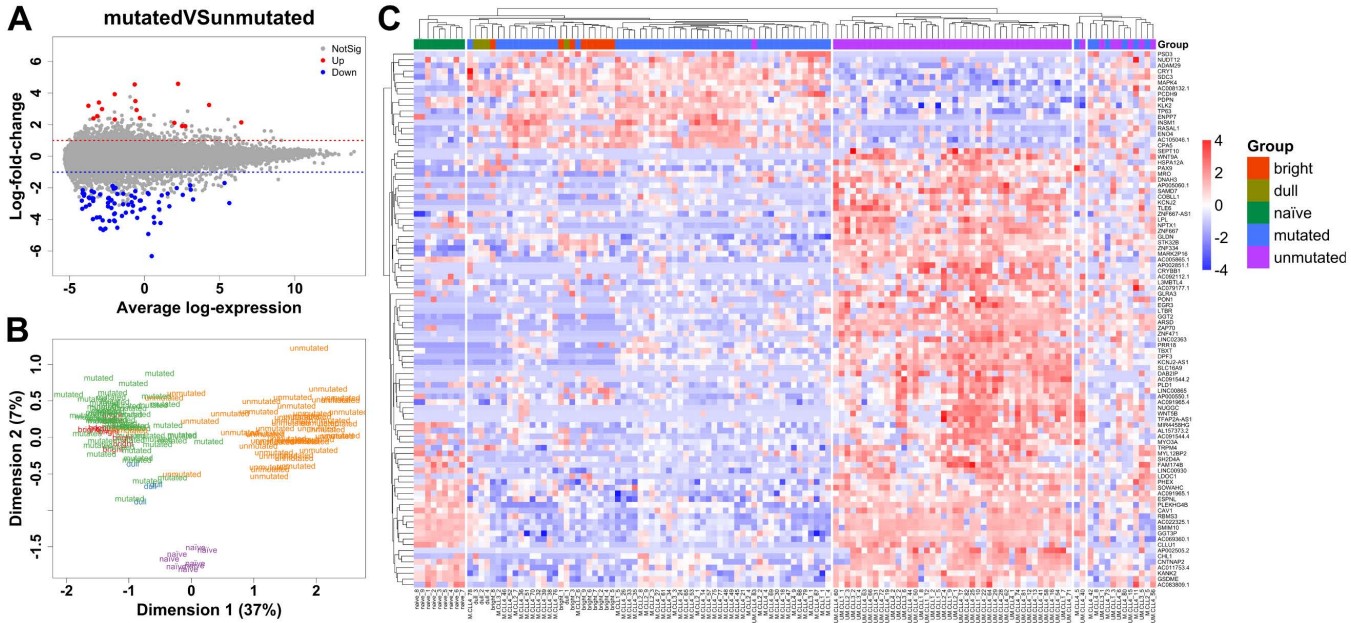

**Fig 2. M-CLL subtype is more closely related to CD27<sup>bright</sup> memory B cells (n = 116). (A)** Mean difference plot (MA plot) showing 94 DEGs, with absolute logFC (LFC > 1), between M- and UM-CLL. Dashed blue and red lines represent LFC at −1 and 1, respectively. **(B)** MDS plot using the same 94 DEGs showing the similarities and differences among the five groups. **(C)** Heatmap for the same 94 DEGs showing the clustering of the five groups.

CD27<sup>bright</sup> and CD27<sup>dull</sup> subgroups cluster together with M-CLL rather than UM-CLL (Fig 2C). This indicates that M-CLL is more transcriptionally similar to CD27<sup>bright</sup> MBCs.

### Pathway analysis to compare the functionality of the M- and UM-CLL to CD27<sup>bright</sup> memory B cells

Next, differential expression analysis was carried out, comparing CLL subgroups to CD27<sup>bright</sup> MBCs (here shortened bright; closest healthy B cell subset) (S3 File). The pairwise comparison bright *vs* mutated shows a total of 391 highly DEGs; 109 upregulated and 282 downregulated DEGs with absolute logFC > 1, respectively (S3 File). The pairwise comparison bright *vs* unmutated shows a total of 471 highly DEGs; 138 upregulated and 333 downregulated DEGs with absolute logFC > 1, respectively (S3 File). Then, pathway analysis was performed using only the significant DEGs (adjusted p-value < 0.05 and absolute logFC > 1). The lists of DEGs for each of the three pairwise comparisons (mutated *vs* unmutated, bright *vs* mutated and bright *vs* unmutated) were used as input for Metascape pathway analysis [27].

The differentially expressed pathways between M- and UM-CLL showed some traditional pathways such as cell proliferation, epithelial cell differentiation, apoptotic and Wnt signaling (Fig 3A and S4 File). Some uncommon pathways between M- and UM-CLL included steroid hormone receptor signaling pathway, cell projection morphogenesis, actin-filament-based movement, cell-cell adhesion, osteoblast differentiation and regulation of cholesterol transport and lipid colocalization.

When comparing CLL subtypes to healthy counterparts of CD27<sup>bright</sup> MBCs, we find shared molecular causes of both CLL subtypes (Fig 3B and S4 File). Some of these pathways are related to T-cell activation, costimulation, immunoglobulin production, cell adhesion molecules as well as neuroactive ligand-receptor interaction. However, some pathways were uniquely altered for M-CLL when compared to CD27<sup>bright</sup> MBCs, such as immune cells' migration, activation and chemotaxis as well as inflammatory response and platelet activation, signaling and aggregation (Fig 3C and S4 File). Also, we find uniquely altered pathways for UM-CLL when compared to CD27<sup>bright</sup> MBCs such as non-canonical NF-KB, leukocyte adhesion, macrophage cell differentiation and cholesterol storage (Fig 3D and S4 File).

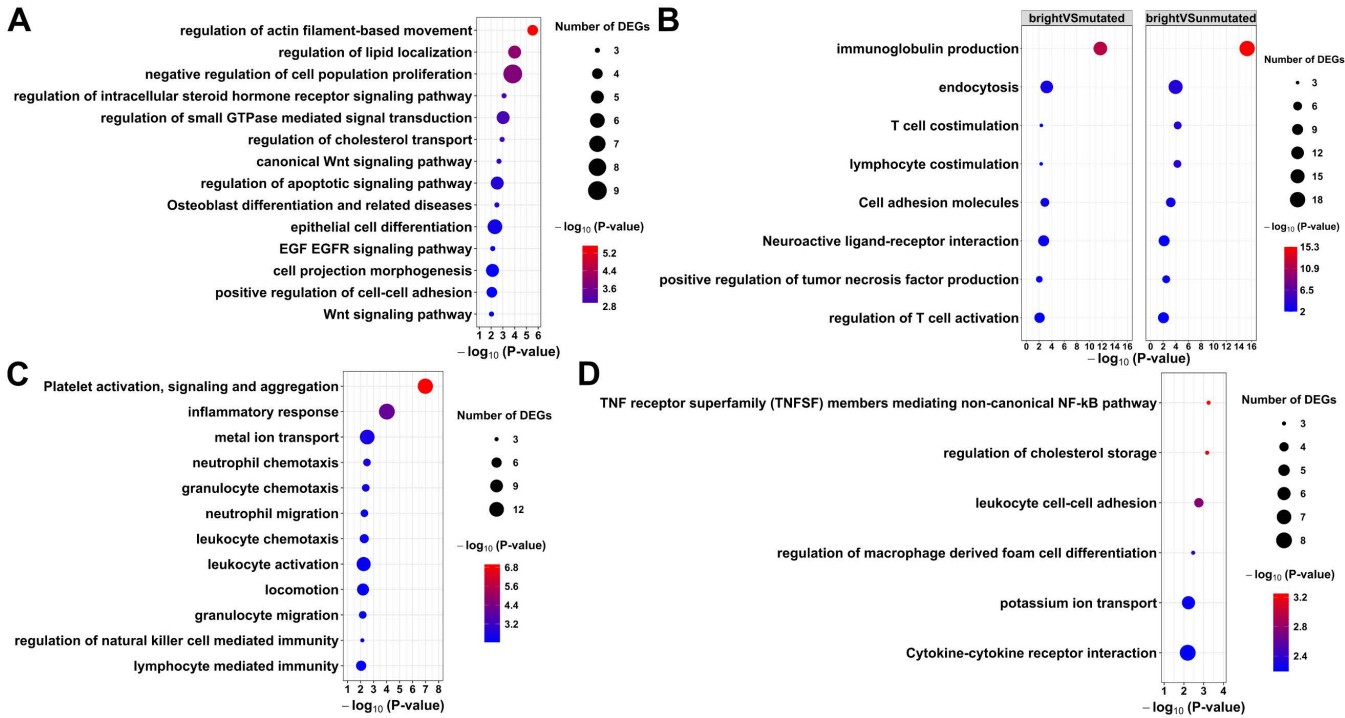

**Fig 3. Metascape pathway analysis for the DEGs from specific pairwise comparisons. (A)** Differentially expressed pathways between M- and UM-CLL. **(B)** Shared enriched pathways between the two pairwise comparisons (M-CLL *vs* CD27bright MBCs and UM-CLL *vs* CD27bright MBCs). Unique enriched pathways for the pairwise comparisons M-CLL *vs* CD27bright MBCs **(C)** and UM-CLL *vs* CD27bright MBCs **(D)**.

## Traditional and potential biomarkers for CLL subtypes

To understand if we find potential biomarkers for CLL subtypes, we investigated the list of significant highly expressed DEGs from the pairwise comparison mutated *vs* unmutated. We found some traditional biomarkers that have been discussed in the literature earlier for distinguishing CLL subtypes such as *ZAP70* and *LPL* (Figs 4A and 4B). *CD27*, although not significantly different between mutated and unmutated subtypes, showed a slight elevation in expression in UM-CLL compared to M-CLL (Fig 4C). *ZNF667* and *ZNF667-AS1* are significantly and highly expressed genes between M- and UM-CLL and fit as potential biomarkers for CLL subtypes (Figs 4D and 4E).

## CLL and GC B cell substages

Our previous data on CD27bright MBCs suggested that they mainly derive from the GC reaction, and therefore we explored the possible GC origin of the two CLL subtypes. We reasoned that it is possible that they derive from different substages of the GC reaction due to the difference in SHM content. We applied the single-cell markers for the GC zones and substages defined by Holmes and colleagues (dark zone, intermediary and light zone GC B cells) using gene set variation analysis (GSVA) [26]. Each GC substage is defined by 100 markers (50 upregulated and 50 downregulated). In total, we have 26 sets of markers for the thirteen GC B cell subtypes, i.e., two sets of markers per GC substage. Differential enrichment score for each GC substage was calculated, using the enrichment scores for each paired upregulated and downregulated sets of markers, for each sample to conclude the highest positive differential enrichment score for each sample, i.e., the dominant GC substage for each sample. We found that a substantial proportion of UM-CLL patients showed a dominant expression profile similar to the dark zone and the early intermediary stage from the GC (Figs 5A and

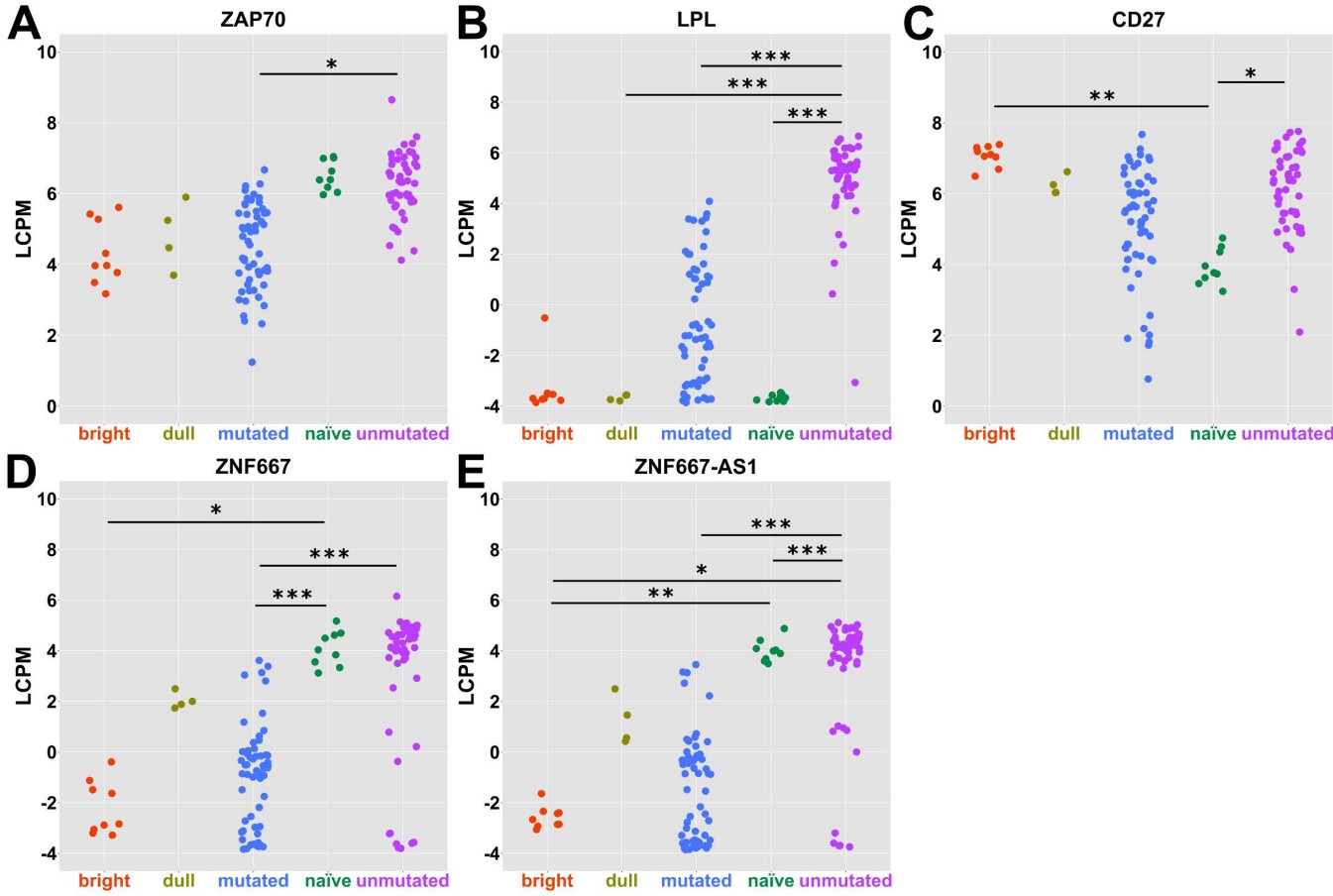

**Fig 4. Traditional and potential biomarkers for CLL subtypes.** Dot plots show the expression levels (log count per million (LCPM)) in each group for selected biomarkers such as *ZAP70* **(A)**, *LPL* **(B)**, *CD27* **(C)**, *ZNF667* **(D)** and *ZNF667-AS1* **(E)**. Empirical Bayes moderated t-test with Benjamini-Hochberg correction was used and p-values are shown as asterisks (* = p ≤ 0.05, ** = p ≤ 0.01 & *** = p ≤ 0.001).

5B, S5 File and S2 Fig). The GC markers also predicted that cells from the M-CLL patients had differentiated further than UM-CLL, and were found mainly similar to a later intermediary stage (Figs 5A and 5B). Our results indicate that both UM- and M-CLL could at least partially derive from the GC reaction with M-CLL coming from a more differentiated GC B cell than the UM-CLL, thus, in line with the higher mutation rate in the former subset.

## Discussion

It is a longstanding controversy whether the two different subtypes of CLL derive from a similar or a different B cell origin [7,10,24]. Our results demonstrate that the transcriptome of M-CLL partially overlaps with the GC-derived and highly mutated CD27[bright] MBCs and also to a higher proportion with pre-MBC from GC stages (Figs 2B, 2C and 5A). This general trend of resemblance, of healthy B cells to M-CLL rather than UM-CLL, could explain why M-CLL is associated with a better prognosis than UM-CLL. Our data also suggests that M-CLL might have passed through more stages in the GC than the UM-CLL (Figs 5A and 5B). This is in line with earlier data where it has been shown that M-CLL could derive from MBCs and that patients with M-CLL, in many cases, have mutations in genes active in the GC, e.g., *KLHL6*, which further supports the hypothesis of a GC-derived origin of M-CLL [4,28,29]. Morabito and colleagues have identified 20 important

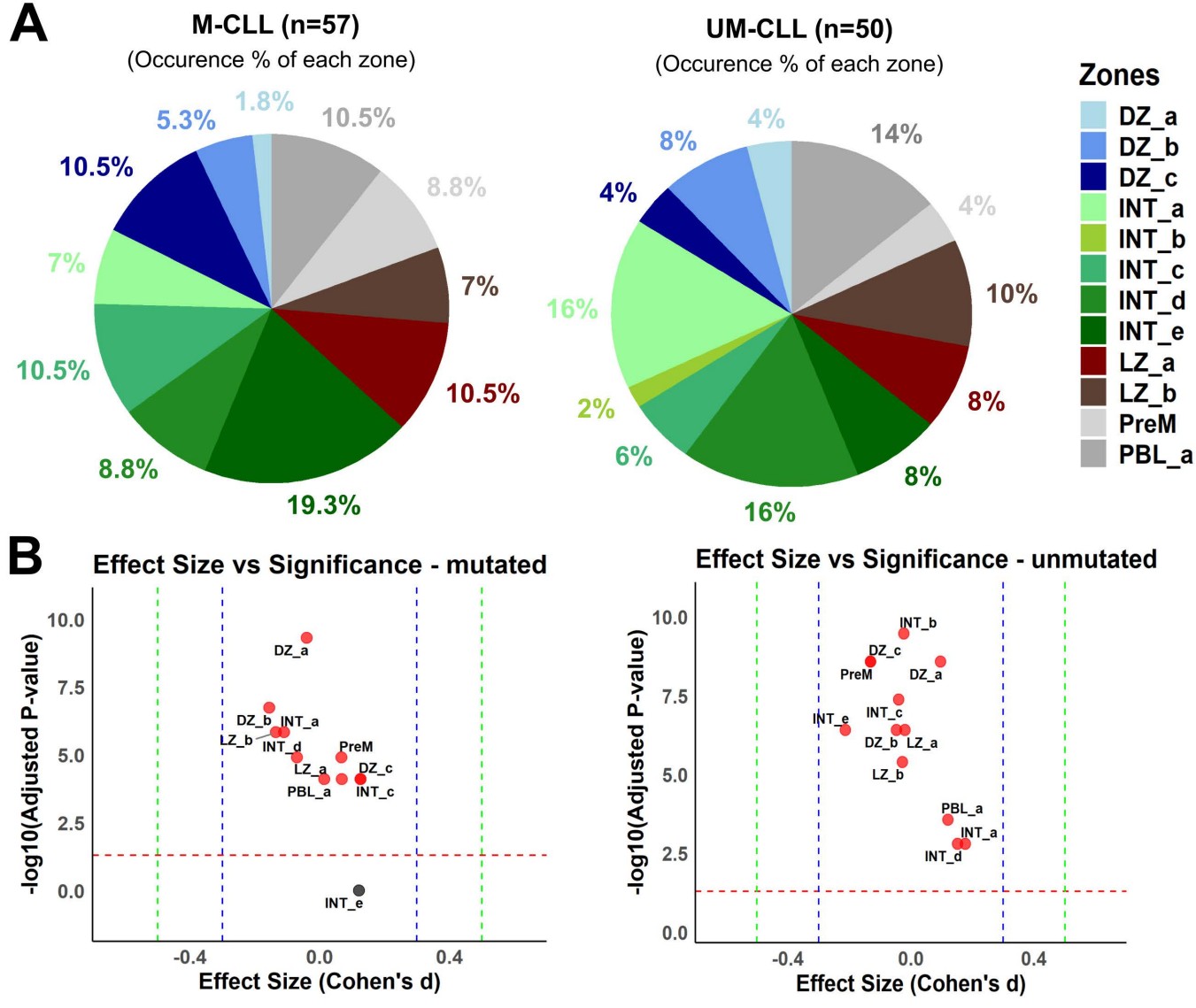

**Fig 5. CLL subtypes and the GC substages (n = 107). (A)** Pie chart showing the percentages of occurrence of each GC zone in CLL subtypes. **(B)** Dot plot shows the significance level and effect size of GC zones in CLL subtypes. Dashed red line at 1.3 represents the significance threshold for -log10 significant adjusted p-value of 0.05. Dashed blue line at 0.3 and dashed green line at 0.5 represent medium size and large size effects, respectively according to Cohen's d. Red dots represent significant GC zones.

genes in CLL subtyping as surrogate markers for IGHV mutational status [30]. In our data, we find 9 of these 20 genes to be significantly and highly deregulated between M- and UM-CLL such as *CLLU1*, *CRY1*, *LPL*, *PHEX*, *SEPT10*, *ZAP70*, *ZNF667*, *ZNF667-AS1* and *COBLL1* (Figs 4A-E and S3 Fig). We propose *LPL*, *ZNF667* and *ZNF667-AS1* as potential markers in CLL subtype stratification.

First, we tried to understand the pathway differences between the two CLL subtypes. We found some interesting pathways that could explain some of the heterogeneity in CLL (Fig 3A). Steroid hormone receptor signaling pathway is altered between M- and UM-CLL which means sex-specific differences in both subtypes (Fig 3A). It has been shown that females exhibit a significantly better overall prognosis with UM-CLL and marginal, non-significant better prognosis with M-CLL [31].

In general, CLL is of higher incidence and is associated with a worse prognosis in men [31]. Skeletal disorders seem to occur with advanced cases of CLL with some evidence of direct correlation between serum tumor necrosis factor-alpha, responsible for stimulated bone resorption, and UM-CLL [32]. It has been suggested that leukemic B cells affect other cells in the tumor microenvironment, which might interfere with physiological bone homeostasis [33,34].

Cell projection morphogenesis, cell-cell adhesion and actin-filament-based movement pathways are altered between M- and UM-CLL (Fig 3A). and have been suggested to be more pronounced in UM-CLL which has more dynamic and aggressive cell projections for motility [35]. This could relate to the dynamics of epithelial mesenchymal transition (EMT) and mesenchymal to epithelial transition (MET) [36]. *S100A4* gene (EMT marker) showed positive correlation with the poor CLL prognosis, possibly UM-CLL, similarly to *ZAP70* [37]. In our data, leukocyte cell-cell adhesion pathway is specifically deregulated for the UM-CLL and not M-CLL, when compared to healthy CD27[bright] MBCs (Fig 3D). *ZNF667* and its anti-sense *ZNF667-AS1* are functionally correlated and differentially expressed between M- and UM-CLL (Figs 4D and 4E) suggesting possible involvement in EMT processes [38]. It is mostly anticipated that their upregulation is associated with lower proliferation, migration and invasion as they act as tumor suppressor genes in some cancers [38–40]. However, in CLL, we see the opposite pattern, similar to glioma [41]. We hypothesize that *ZNF667* and its anti-sense-1 are tumor promotor genes in CLL that lead to a worse prognosis. Very few studies discussed their roles in CLL as potential prognostic markers and their association with UM-CLL as well as their contribution to worse prognosis [30,42–44].

The role of cholesterol in CLL pathogenesis and treatment has been discussed by some studies [45–48]. However, only two studies have discussed the different impacts of cholesterol on the two CLL subtypes [49,50]. They argued that cholesterol is impacting UM-CLL more than M-CLL [49,50]. Similarly, we find regulation of cholesterol storage pathway is specifically deregulated for the UM-CLL, and not M-CLL, when compared to healthy CD27[bright] MBCs (Fig 3D). *LPL* shows higher expression in UM-CLL compared to M-CLL (Fig 4B) hence, possibly creating different metabolic profiles for cholesterol in CLL subtypes [51]. This is consistent with previous findings that UM-CLL showed higher uptake and metabolism of cholesterol and lipid [47,52]. This lipid profile switching could be one of the reasons behind the aggressive nature of UM-CLL.

An interesting pathway shared by both CLL subtypes is neuroactive ligand-receptor interaction (Fig 3B). CLL rarely involves the Central Nervous System (CNS) in a clinically significant manner and that symptomatic CNS involvement occurs approximately in 1% of CLL patients [53–55]. However, autopsy studies suggest that the involvement of CNS might be more common with a prevalence reported ranging from 7 to 71% [55,56]. The nervous system involvement (peripheral or central) can also go unnoticed due to the masking effects of the medications used in CLL treatment [57]. It has not been further discussed which CLL subtype is more involved with the nervous system. The role of B cells has been discussed earlier in the pathogenesis of various autoimmune and neurodegenerative disorders [58–62]. Cytokine production, secretion and interaction are some of the suggested mechanisms through which B cells can exert influence on the CNS [58,59,63]. Here, we exclusively find that cytokine-cytokine receptor interaction pathway is specifically altered in UM-CLL, not M-CLL, when compared to healthy CD27[bright] MBCs (Fig 3D). This piques the topic of whether CNS is more impacted in UM-CLL subtype or not.

Platelet activation, signaling and aggregation pathway is specifically deregulated for the M-CLL, and not UM-CLL, when compared to healthy CD27[bright] MBCs (Fig 3C). This pathway can be altered as a collateral damage from other faulty lymphoid cells such as B cells and Natural Killer (NK) cells which are targeted in CLL [64]. The alterations in the metal ion transport and locomotion pathways can also contribute to the deregulation of platelet activation and coagulation process [65,66]. Neutrophils also play a role in this process and can affect platelets and clotting significantly [67,68]. Changes in platelets and bleeding in general can also occur as a response to anti-cancer treatments for CLL such as ibrutinib [69]. However, the association of this pathway to a certain subtype of CLL has not been discussed.

Gaidano and colleagues proposed that UM-CLL could come from a MBC origin, and this is supported by our results which indicate that UM-CLL shows transcriptional similarity to an early GC origin with limited LZ experience (Figs 5A and 5B) [70]. However, we cannot exclude that it would come from a T-cell or GC-independent activation pathway as suggested previously

[8,9]. Thus, not a clear MBC origin, but possibly a pre-MBC stage and this goes hand in hand with the fact that both pre-MBC stage and UM-CLL have low SHM rate and higher proliferative capacity [71]. Our results suggest that UM-CLL is less transcriptionally similar to the healthy B cell subsets, thus probably not have matured totally into a memory B cell (Figs 2B and 2C). Seifert and colleagues suggested that CD5+ B cells are the origin of both UM- and M-CLL [24]. Interestingly, our results could well support that UM-CLL can derive from non-MBC (CD5+CD27-) origin that enters the GC where it gets activated and transforms at an early stage; hence not naïve B cell anymore, but rather an activated naïve B cell that entered the GC (Figs 5A and 5B). Additionally, activated B cells may upregulate CD5 which can protect them from apoptosis [24], but also could be a possible transformation to UM-CLL. A preprint from Seifert and collaborators indicates that both UM-CLL and M-CLL might derive from the GC reaction and their single-cell BCR sequencing analysis suggests that the 5% most mutated B cells in M- and UM-CLL are similar in their SHM frequency in their Ig V genes [72]. In the same study, they also observed a third subset of CLL cells which is present in both UM-CLL and M-CLL cases and that these cells showed similar proliferative capacity as CD5+ MBCs. This could be one reason why we observe a lot of shared pathways in both M- and UM-CLL (Fig 3B).

There are some limitations to this study. We have used tonsillar GC substages as a reference for CLL subtypes. There are differences across the lymph nodes and some niche-specific signals may not be present in circulating cells. However, the core transcriptional program remains conserved for B cell states during GC maturation. Interpretations of the mapping results of CLL samples to GC substages should be carefully considered in the transcriptional context only. Some of the observed patterns could occur due to aberrant transcriptional programs in CLL, such as constitutive NF-κB activation, BCR signaling, or defective apoptotic regulation, rather than signatures for GC substages. Thus, the results could represent the relative positioning of CLL transcription profiles along GC-like transcriptional continuum rather than representing true GC substages in CLL. This is a transcriptomics-focused study and no validations on the protein or function level have been carried out.

In conclusion, M-CLL shows the closest transcriptional similarity to CD27<sup>bright</sup> MBCs whereas UM-CLL expresses a transcriptome that is more different from other B cell subsets. However, UM-CLL shows a transcriptional trend toward an early intermediary stage of GC B cells while M-CLL shows a trend toward a later stage of the GC B cells which could potentially explain the IGHV mutational status of M-CLL. We suggest that *LPL*, *ZNF667* and *ZNF667-AS1* could be used as biomarkers for CLL patient stratification as well as mechanistic regulators of CLL pathology through cholesterol and EMT regulation.

## Materials and methods

### Ethics statement

The RNA sequencing data from anonymous healthy donors' buffy coats used in this study was initially produced within the scope of a different study (Pour Akaber et al, unpublished data with GSE307095, recruitment period from 8th of January 2019–2nd of March 2022) which has been approved by the ethical committee of Gothenburg region (no 727−17). Written consent was not required for these samples as no personal information or identity was recorded (Swedish law 2003: 460, paragraphs 4 and 13). The study was conducted in accordance with the Helsinki Declaration.

### Human samples

All materials from healthy adults collected in this study were from anonymous buffy coats or from previously published studies. For the RNA sequencing data from our previously published dataset (the first healthy dataset herein) sex and age were known, see [25]. The RNA sequencing data from anonymous healthy donors' buffy coats used in this study was initially produced within the scope of a different study (Pour Akaber et al, unpublished data, GSE307095).

### B cell isolation and cell sorting

Heparinized peripheral blood mononuclear cells (PBMCs) were isolated by Ficoll Paque Plus 206 (Amersham Pharmacia-Biotech) density-gradient centrifugation. Buffy coats were incubated with RosetteSep human B cell enrichment antibody

cocktail (Stemcell Technologies) and then B cells were isolated by density-gradient centrifugation. B cells were then stained with antibodies against CD19, CD24, CD27 and CD38 (BD Biosciences). The B-cell subsets were gated as follows: mature-naïve B cells as CD24+CD27-CD38+ and CD27bright MBCs as CD24+CD27+. Sorting was performed using the Sony SH800 cell sorter (Sony Technologies) or BD FACSAria Fusion (BD Biosciences). Sort purities were >98%.

## RNA extraction and RNA sequencing

Total RNA was isolated from sorted B cells of five healthy donors from anonymous buffy coats using RNeasy Plus Micro kit (Qiagen). The quantity of RNA extracted was measured using the Qubit fluorometric quantification system (Thermo Fischer Scientific). The TruSeq RNA Sample Preparation v2 Kit (Illumina) was used to isolate polyadenylated mRNA with oligo-dT beads, second-strand cDNA synthesis and NGS library preparation. During the second strand synthesis, dUTP was incorporated in place of dTTP, thus preventing the amplification of this strand during the subsequent PCR step and retaining strand information. Unique indexes, included in the standard TruSeq kit set A (Illumina) were incorporated at the adaptor ligation step for multiplex sample loading on the flow cells. The resulting constructs were purified by two consecutive AMPure XP beads cleanup steps and enriched by 15 cycles of PCR. The quality and quantity of the libraries were assessed using an Agilent High Sensitivity D1000 ScreenTape (Agilent Technologies) and Qubit dsDNA HS Assay Kit (Thermo Fischer Scientific), respectively. Twelve libraries were pooled in equimolar amounts and loaded at 1.2 pM onto the flow cell (High Output Kit v 2.5 150 cycles, Illumina). The sequencing run was performed in paired-end mode (2 X 75-bp reads) using the NextSeq 500 platform. Base-calling was performed by the instrument computer using Illumina Real Time Analysis (RTA) software that is integrated with NextSeq Control Software (NCS) and provides a summary of quality statistics as per Illumina's acceptance criteria for sequencing. CASAVA 1.8.2 was used for de-multiplexing and converting base calls to paired-end FASTQ files. Data from the second healthy dataset (Pour Akaber et al) was accessed for analysis on 30th of January 2020.

## RNA sequencing bioinformatic analysis

**Differential expression and pathway analysis.** Herein, four publicly available CLL datasets were included [16,26,73] containing transcriptomic data from both UM- and M-CLL subtypes, and one dataset (first healthy dataset herein) from healthy donors containing three subsets (naïve B cells, CD27dull MBCs and CD27bright MBCs) from four healthy individuals [25] (S1 File). In addition, a new dataset, obtained from RNA sequencing of five healthy donors, was included as mentioned above. Raw data have been controlled, aligned and quantified using the nf-core/rnaseq pipeline (v3.2) with the Ensembl human reference genome GRCH38/hg38 – Ensembl release 96 [74] (S6 File). R programming language was used for the downstream analysis. The session information includes the packages and the versions used in the analysis (S7 File). Outliers were detected with OUTRIDER and samples showing more than 2 genes in the list were considered outliers. 16 samples were considered outliers, and removed [75] (S2 File). Batch correction for all the datasets was performed based on the origin of the dataset and the Run type (paired or single end). All datasets were integrated. Filtration, normalization and quality control were performed. MDS plots were created with plotMDS function from limma with Euclidean distance on leading log-fold changes as a distance measure. Differential expression analysis was performed using limma and edgeR pipeline with voomWithQualityWeights function to include sample-level weights [76]. The lists of differentially expressed genes (DEGs) were obtained for the different pairwise comparisons between the groups (with an adjusted p-value/False Discovery Rate of 0.05 and absolute log fold-change cutoff 1 using "treat" function). Pathway analysis was performed through Metascape, by inputting the lists of DEGs for the separate pairwise comparisons [27] (accessed on 12.06.2024). We considered only the significantly altered pathways with adjusted p-value <0.05.

**CLL markers analysis for GC B cell substages.** Holmes, *et al* have suggested sets of markers for different substages of the GC (based on their location) by analyzing single-cell transcriptomics of the tonsillar GC B cells of

healthy donors. They demonstrated that subgroups of diffuse large B cell lymphoma could be identified using their defined markers. Therefore, their list for the up- and downregulated markers for the thirteen substages in the GC were downloaded [26]. Each substage is defined by 100 RNA markers (50 markers upregulated & 50 markers downregulated). In total, 26 sets of markers for thirteen GC B cell subtypes were used.

Gene set variation analysis (GSVA) was performed, using GSVA R package, to measure how much each set of markers was enriched for these thirteen GC substages in our healthy and CLL subjects [77]. A higher positive enrichment meant a stronger upregulation of a set in a sequenced sample, while a lower negative score meant stronger downregulation. Then, to virtually map each subject in the GC, a differential score for each substage for each sample was calculated by subtracting the enrichment score for the downregulated markers from the score of the upregulated ones. The highest positive differential enrichment score (representative for both direction of regulation and enrichment) for each sample decided the dominant GC B cell subtype for each sample. Fischer's exact test was used to check the significantly dominant GC B cell subtypes in each group. P-values have been adjusted (Benjamini-Hochberg) for multiple comparisons to check the significantly dominant zones in each group. Cohen's effect size was implemented as a combination metric with p-values to highlight the dominant zones in an unbiased manner differentiating between trivial and meaningful differences [78] (S5 File). Cohen's D shows the difference between two means relative to the pooled standard deviation of the observations. Effect size is classified as small ($|d| \leq 0.3$), medium ($|d| > 0.3$ & $\leq 0.5$) or large ($|d| \geq 0.5$) according to Cohen [79].

## Supporting information

**S1 Fig. FACS plots.** FACS plots (pre-sort) show the sorting strategy used to separate CD24intCD27-CD38int naive B cells from CD24highCD27bright MBCs. The post-sort purity for the two populations is shown in the lower panels. (TIF)

**S2 Fig. Healthy B cell subtypes and GC substages.** (A) Pie chart showing the percentage of occurrence of each GC zone in B cell subtypes. (B) Dot plot shows the significance land effect size of GZ zones in B cell subtypes. Dashed red line at 1.3 represents the significance threshold for -log10 significant adjusted p-value of 0.05. Dashed blue line at 0.3 and dashed green line at 0.5 represent medium and large size effect respectively according to Cohen. Red dots represent significant GC zones. (TIF)

**S3 Fig. Traditional biomarkers for CLL subtypes.** Dot plots show the expression levels (log count per million (LCPM)) in each group for certain biomarkers such as CLLU1 (A), COBLL1 (B), CRY1 (C), PHEX (D) and SEPT10 (E). P-values are shown as asterisks (* = $p \leq 0.05$, ** = $p \leq 0.01$ & *** = $p \leq 0.001$). (TIF)

**S1 File. Data description.** Naming and description of the used datasets as well as their accession numbers. (XLSX)

**S2 File. Outlier summary.** OUTRIDER significant results with outlier samples. (XLSX)

**S3 File. Differential expression results.** Lists of DEGs for each pairwise comparisons (logFC > 1 or < −1). (XLSX)

**S4 File. Pathway results.** Lists of deregulated Metascape pathways for the pairwise comparisons (mutated *vs* unmutated, bright *vs* mutated and bright *vs* unmutated). (XLSX)

**S5 File. Test results for GC zones.** Results for Fisher's exact test and Cohen's effect size for dominant GC zones in each group.
(XLSX)

**S6 File. Software details.** Nextflow nf-core/rnaseq pipeline and software versions.
(XLSX)

**S7 File. Details of R packages.** R session info for the analysis showing details on used packages and their versions.
(TXT)

**S8 File. Inclusivity-in-global-research-questionnaire.**
(DOCX)

## Acknowledgments

We thank Professor Per-Ola Andersson for his critical review of this manuscript. Thanks to the Accelerating Research in Genomic Oncology-International Cancer Genome Consortium (ARGO-ICGC) for granting access to one of the CLL datasets used in the analysis (Study ID: EGAS00001000374 & Dataset ID: EGAD00001000258).

## Author contributions

**Conceptualization:** Ahmed Mohamed, Andreas Tilevik, Ola Grimsholm.

**Data curation:** Ahmed Mohamed, Ola Grimsholm.

**Formal analysis:** Ahmed Mohamed.

**Funding acquisition:** Andreas Tilevik, Ola Grimsholm.

**Investigation:** Ahmed Mohamed, Sabina Barresi, Andreas Tilevik, Ola Grimsholm.

**Methodology:** Ahmed Mohamed, Luca Giudice, Andreas Tilevik, Ola Grimsholm.

**Project administration:** Ahmed Mohamed, Ola Grimsholm.

**Resources:** Ola Grimsholm.

**Software:** Ahmed Mohamed, Luca Giudice, José Basílio, Andreas Tilevik.

**Supervision:** Luca Giudice, José Basílio, Tarja Malm, Andreas Tilevik, Ola Grimsholm.

**Validation:** Ahmed Mohamed.

**Visualization:** Ahmed Mohamed.

**Writing – original draft:** Ahmed Mohamed, Ola Grimsholm.

**Writing – review & editing:** Luca Giudice, José Basílio, Sabina Barresi, Pradeep Kumar Kopparapu, Vanda Friman, Marco Tartaglia, Tarja Malm, Andreas Tilevik.

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
