## [Decision Letter · Decision Letter 0]

23 Jul 2025

Dear Dr. Mohamed,

Thank you for submitting your manuscript to PLOS ONE. After careful consideration, we feel that it has merit but does not fully meet PLOS ONE’s publication criteria as it currently stands. Therefore, we invite you to submit a revised version of the manuscript that addresses the points raised during the review process by both Reviewers.

We look forward to receiving your revised manuscript.

Kind regards,

Francesco Bertolini, MD, PhD

Academic Editor

PLOS ONE

Journal Requirements:

“OG: Assar Gabrielsson foundation, Anna-Lisa and Bror Björnsson foundation, Adlerbert Research Foundation, the Royal Sciences and Arts in Gothenburg and the

Austrian Science Fund (FWF), project PAT3959823.

MT: Associazione Italiana per la Ricerca sul Cancro (IG28768).

AM: European Union’s Horizon 2020 research and innovation programme under the Marie Skłodowska-Curie agreement No101034307”

5. In the online submission form, you indicated that your data will be submitted to a repository upon acceptance.  We strongly recommend all authors deposit their data before acceptance, as the process can be lengthy and hold up publication timelines. Please note that, though access restrictions are acceptable now, your entire minimal  dataset will need to be made freely accessible if your manuscript is accepted for publication. This policy applies to all data except where public deposition would breach compliance with the protocol approved by your research ethics board. If you are unable to adhere to our open data policy, please kindly revise your statement to explain your reasoning and we will seek the editor's input on an exemption.

Reviewers' comments:

Reviewer's Responses to Questions

**Comments to the Author**

1. Is the manuscript technically sound, and do the data support the conclusions?

Reviewer #1: Yes

Reviewer #2: Yes

2. Has the statistical analysis been performed appropriately and rigorously?

Reviewer #1: Yes

Reviewer #2: Yes

3. Have the authors made all data underlying the findings in their manuscript fully available?

Reviewer #1: Yes

Reviewer #2: Yes

4. Is the manuscript presented in an intelligible fashion and written in standard English?

Reviewer #1: Yes

Reviewer #2: Yes

Reviewer #1: This study offers a novel investigation into CLL subtype origins using germinal center trajectory mapping. The concept is compelling and well-executed, but methodological caveats require clarification to solidify conclusions.

The application of tonsillar GC B-cell markers (Holmes et al.) to infer developmental states in peripheral blood-derived CLL cells presents biological disconnects:

• Tonsillar GC substages rely on niche-specific signals (e.g., CXCL12/CD40L) absent in circulation, risking misinterpretation of transcriptomic signatures as active GC maturation.

• Protein-level validation (e.g., CXCR4/CD83 flow cytometry) and comparison to lymph node/bone marrow microenvironments are lacking.

Could the observed substage signatures reflect aberrant transcriptional mimicry (e.g., constitutive NF-κB) rather than true GC maturation? Please temper claims and discuss limitations.

Additionally, please Revise Methods to passive voice for objectivity.

Reviewer #2: In the manuscript, the authors have tried

1) To investigate the potential cellular origin of M- and UM-CLL subtypes using multiple CLL transcriptomic cohorts downloaded from public databases and compared the data RNA sequencing data generated in their lab from mature-naïve (CD24+CD38int) B cells, CD27dull MBCs (CD24hiCD27dull) and CD27bright MBCs (CD24hiCD27bright).

2) To decipher commonalities and differences between M- and UM-CLL subtypes against healthy B cells through a transcriptomic meta-analysis and,

3) To investigate the potential GC origin of the two CLL subtypes using single-cell data already available.

Specific Comments:

The manuscript title 'Germinal center trajectories and transcriptional signatures define CLL subtypes and their functional regulators,' suggests the inclusion of functional assays. However, based on the manuscript's content, the proposed pathways and gene signature relationships appear to be derived solely from in silico pathway analysis and the available literature. To accurately reflect the scope of the work, it is recommended to revise the title particularly regarding the 'functional regulators' aspect.

The same implies in the abstract also. The authors have mentioned that functional analysis revealed that LPL…..EMT regulation. The term functional analysis should be replaced with functional enrichment analysis.

The abstract needs to clearly outline the methodology used in the manuscript. It needs to include details like number of patients studied, methodology like meta analysis was conducted.

The transcriptional signatures highlighting the traditional and potential biomarkers for CLL sub- types is diverting the focus of the manuscript. Many published papers are already available for transcriptional signatures between these subtypes (Mutated vs unmutated). Hence, the abstract and discussion should be modified accordingly.

Given that the results are based on bioinformatics meta-analysis without functional validation, avoid making very strong conclusions. Please use mild words like 'suggested' or 'hypothesized' in both the abstract and discussion. A statement acknowledging that these findings require functional validation may be added somewhere in the discussion.

The effort should be to make it concise. In the introduction section, paragraph 2 may be more concise. The discussion also needs to be modified and concise in line with the suggestions provided above.

**Do you want your identity to be public for this peer review?** For information about this choice, including consent withdrawal, please see our Privacy Policy

Reviewer #1: **Yes: ** Amir Abbas Navidinia

Reviewer #2: **Yes: ** Ritu Gupta

---

## [Author Response · Author response to Decision Letter 1]

10 Sep 2025

Rebuttal letter (Response to Reviewers), Revised Manuscript with Track Changes and Manuscript have been uploaded in this revision. Some general comments on the decision letter:

1- Files and figures have been renamed and uploaded to PACE to match PLOS ONE's style requirements.

2- PLOS’ questionnaire on inclusivity in global research is filled and attached with the submission as "Supporting Information".

3- Financial disclosure and role of funders have been updated in the cover letter of the revision.

4- Data restriction has been updated in the manuscript and submission portal. All data necessary to replicate the study are available open access. Only the raw data (fastq files) of one cohort is uploaded to NCBI, but not open access yet and it will be open access after the acceptance of this manuscript.

5- Processed data and analysis pipeline are now openly available on Zenodo with the DOI number provided in the manuscript.

6- Ethics statement is now only mentioned in the methods section.

7- Citations and reference list have not been modified.

8- Comments from the editor and reviewers have been incorporated into the revised manuscript and addressed separately in the "response to reviewers" file.

9- We accept the publishing of the review process.

---

## [Decision Letter · Decision Letter 1]

6 Oct 2025

Germinal center trajectories and transcriptional signatures define CLL subtypes and their pathway regulators

PONE-D-25-22015R1

Dear Dr. Mohamed,

We’re pleased to inform you that your manuscript has been judged scientifically suitable for publication and will be formally accepted for publication once it meets all outstanding technical requirements.

Kind regards,

Francesco Bertolini, MD, PhD

Academic Editor

PLOS ONE

Additional Editor Comments (optional):

Reviewers' comments:

Reviewer's Responses to Questions

**Comments to the Author**

Reviewer #1: All comments have been addressed

Reviewer #2: All comments have been addressed

2. Is the manuscript technically sound, and do the data support the conclusions?

Reviewer #1: Yes

Reviewer #2: Yes

3. Has the statistical analysis been performed appropriately and rigorously?

Reviewer #1: Yes

Reviewer #2: (No Response)

4. Have the authors made all data underlying the findings in their manuscript fully available?

Reviewer #1: Yes

Reviewer #2: (No Response)

5. Is the manuscript presented in an intelligible fashion and written in standard English?

Reviewer #1: Yes

Reviewer #2: Yes

Reviewer #1: (No Response)

Reviewer #2: (No Response)

**Do you want your identity to be public for this peer review?** For information about this choice, including consent withdrawal, please see our Privacy Policy

Reviewer #1: **Yes: ** Amir Abbas Navidinia

Reviewer #2: **Yes: ** Ritu Gupta

---

## [Editor Report · Acceptance letter]

PONE-D-25-22015R1

PLOS ONE

Dear Dr. Mohamed,

I'm pleased to inform you that your manuscript has been deemed suitable for publication in PLOS ONE. Congratulations! Your manuscript is now being handed over to our production team.

Kind regards,

on behalf of

Dr. Francesco Bertolini

Academic Editor

PLOS ONE